# Food-Based Dietary Guidelines for Seafood Do Not Translate into Increased Long-Chain Omega-3 Levels in the Diet for U.S. Consumers

**DOI:** 10.3390/foods10081816

**Published:** 2021-08-05

**Authors:** Michael F. Tlusty

**Affiliations:** School for the Environment, University of Massachusetts Boston, Boston, MA 02125, USA; michael.tlusty@umb.edu

**Keywords:** food-based dietary guidelines, seafood consumption, long-chain omega-3 fatty acids

## Abstract

Humans under-consume fish, especially species high in long-chain omega-3 fatty acids. Food-based dietary guidelines are one means for nations to encourage the consumption of healthy, nutritious food. Here, associations between dietary omega-3 consumption and food-based dietary guidelines, gross domestic product, the ranked price of fish, and the proportions of marine fish available at a national level were assessed. Minor associations were found between consumption and variables, except for food-based dietary guidelines, where calling out seafood in FBDGs did not associate with greater consumption. This relationship was explored for consumers in the United States, and it was observed that the predominant seafood they ate, shrimp, resulted in little benefit for dietary omega-3 consumption. Seafood is listed under the protein category in the U.S. Dietary Guidelines, and aggregating seafood under this category may limit a more complete understanding of its nutrient benefits beyond protein.

## 1. Introduction

The context of fish (defined by the FAO (2020) as “fish, crustaceans, molluscs and other aquatic animals, but excludes aquatic mammals, reptiles, seaweeds and other aquatic plants”) consumption varies across the globe. For many, the consumption of aquatic-produced foods is a function of the accessibility to fisheries, along with socio-economic dependence on those fisheries (both inland and oceanic), and the prevalence of aquatic-produced foods within the local food system [1]. For these often-low-income countries, fish is a necessity, because it can contribute to >20% of the animal protein supply [1,2], prevents micronutrient deficiencies [3]; thus, substituting fish is not realistic [4], and increasing consumption would be beneficial. For other countries, particularly those in the high-income category, the context is that terrestrial-based proteins are more abundant, and fish is not a necessary protein source [5]. Here, a greater variety of choice and availability of nutrients leads to the question of substitution. Consumers often pose the “instead of” question [6], and consider “*Do I have the fish, or instead do I have some other animal source food*?”. Yet is micronutrients are deficient, then substituting a good source of micronutrients for a poor source is not a viable option. Furthermore, within high-income countries, people are still hungry, although for many of those that can obtain sufficient calories, their dietary choices lead to where they do not consume a full complement of recommended nutrients [7]. Across all economic levels, it is important that suggested nutritional substitutions be within the proper and intended purpose for achieving nutritional sufficiency. Here, the concept that fish consumption in high-income countries is more a concern about nutrients rather than protein [8,9] is explored. Specifically, the focus is on the case of long-chain omega-3 fatty acids (LCn-3s) within the United States (USA).

LCn-3s are important nutrients for humans because they have significant cardiovascular, neurocognitive, and psychological health benefits [10,11,12,13]. Marine oily fish are considered the primary source of LCn-3 in the human diet [14,15], because LCn-3s originate in marine algae, and are bioaccumulated up food chains. There are some terrestrial sources of LCn-3s, as well as human populations that have an increased ability to elongate short omega-3 to LCn-3 [16]. In the United States, fish and seafood constitute 71% of the diets of citizens who consume a high amount of LCn-3s [17]. LCn-3s are commonly harvested from finite fisheries [18], and a global gap between supply and demand for this nutrient [18,19] makes them limited in many diets globally [20]. Most national dietary guidelines that have a daily recommended nutritional intake of LCn-3 place the value at ≥250 mg per day for adults [14]; however, global availability is only 149 mg EPA+DHA per capita daily [18]. Inequity in distribution and availability [21,22] are likely to exacerbate the number of people with limited dietary LCn-3. 

Food-based dietary guidelines (FBDGs) are used to “provide advice on foods, food groups and dietary patterns to provide the required nutrients to the general public to promote overall health and prevent chronic diseases” [23]. Given that LCn-3s are limited globally [18], it is not surprising that LCn-3 from seafood (note the change from ‘fish’ in the first paragraph to ‘seafood’ here, due to the marine origin of LCn-3s) is limited in diets [20]. However, it is unknown whether FBDGs to consume seafood are linked to an increased consumption of LCn-3s from seafood. With the limited nature of this critical nutrient, the FBDGs can be viewed as a good place to encourage the increased consumption of limited nutrients.

This review threads a link between disparate but complementary ideas and data sets. The following are known separate data points for global seafood:-How many countries include “seafood” or “fish” in their FBDGs [24,25,26,27];-The amount of fish available within each country [28];-The calculated level of dietary LCn-3s [20];-The country-level variations in fish prices [29].

However, it is not known if there is a correlation between FBDGs for “seafood” and dietary LCn-3s, how this relationship may be moderated by the amount of fish available nationally, or its cost. 

Seafood consumption in the United States has been extensively discussed [9,30]. As demonstrated globally, multiple lines of disparate but complimentary data exist but lack cohesiveness. The following facts are available for seafood in the United States:-U.S. citizens under-consume fish based on NHANES data analysis [9,30], a fact so obvious it is stated in the Dietary Guidelines for Americans [31];-The type of fish consumed matters with respect to LCn-3 [32], with a call for Americans to include oily fish [31], or more specifically, salmon [33];-Americans do not consume all fish species equally, and shrimp, salmon, and canned tuna comprise 50% of their annual seafood consumption [34]; however, only salmon contains enough LCn-3 to meet a recommended intake of 250 to 500 mg per day [35];-Consuming oily fish multiple times per week along with ingesting supplements results in the highest index of the percentage of LCn-3 as a percentage of total erythrocyte fatty acids [36].

However, after two decades of calls to increase LCn-3 intake [37], are American dietary patterns changing to increase the dietary availability of LCn-3? 

Understanding influences on LCn-3 consumption is critical to provide the link between FBDGs for seafood and the amount consumed, and how the guidelines may or may not be followed, will help identify additional steps for modifying FBDGs for increasing dietary benefits.

## 2. Materials and Methods

Here, the published data sets on the inclusion of the recommendation to eat “seafood” (often referred to generically in FBDGs as “fish”) within FBDGs [27] are compared to country-level seafood availability [28], as well as the dietary inclusion of seafood-based LCn-3s [20]. These are also compared to the average price for seafood at a national level [29], and the per capita gross domestic product [38]. Diet data are subject to overestimation [39,40,41], and as such, these disparate data sets were analyzed with a ranked trend analysis. A quadrate trend analysis was conducted, where data for each country were identified as being below or above a median value for that variable; then, two variables assessed within a 2-factoral space. If the dominant cells indicated that both variables were below or above the median, this indicated a positive association. Factors trending negatively had below-above median associations. Six total comparisons were made; therefore, a protected *p*-value of 0.008 (0.05/6) was used to prevent reporting spurious correlations.

The translation of the FBDG for a singular country, the United States, was then explored through an examination of seafood consumption patterns [34]. The industry-calculated seafood consumption patterns were compared to their ability to meet a guideline of 8 oz (226.7 g) per week (or 32.4 g/d, [31]) for a 2000-calorie daily diet. The LCn-3 levels of EPA and DHA [42] of the top ten seafood species consumed in the United States [34] were used to assess cumulative LCn-3 consumption through these species for 2019 data. The top ten list accounted for only 74.8% of all seafood consumed; therefore, the remaining 25.2% was assumed to have an LCn-3 level equal to the weighted average of the top 10 species consumed. The top ten consumption patterns were then extended so the total amount consumed was the recommended (8 oz per week) total. The LCn-3 consumption was then recalculated based on whether consumer behavior was equivalent to the recommended level.

## 3. Results

A number of prior studies have examined the details within different FBDGs, and found that seafood (or fish) was mentioned in 58% [24], 80% [25], 83.5% [27], or 83.7% [26]. A smaller number of FBDGs provided information on either the serving size or the number of meals per week. Graham et al. [26] found that 35 FBDGs reported servings per week, and Herforth et al. [24] reported an average of 2.2 servings per week, but only 23 provided the serving size. It is difficult to generalize serving size given the wide variability in how this metric is reported. However, Springmann et al. [27] were able to deduce a serving size of 32.9 (±17.4 g/d, mean ±1. s.d., range 4.6–93.5, *n* = 65) for 90% of the FBDGs that suggested eating seafood. 

The median quadrant analysis demonstrated an overall positive trend (**χ**^2^ = 11.0029, *p* < 0.001, Table 1) between the FBDG serving size, determined by Springmann et al. [27], and dietary LCn-3, determined by Micha et al. [20] (Figure 1, interactive data available at https://www.tlusty.solutions/n3.html (accessed on 28 June 2021). Of factors that may influence dietary LCn-3, the presence of a seafood recommendation in the FBDGs, per capita GDP, the total fish available in the country, and the percentage of fish from marine sources all showed a positive association with the dietary LCn-3 (for all values, **χ**^2^ > 9.7, *p* < 0.002, Table 1). The ranked price per fish in various countries was negatively associated with the amount of LCn-3s in the diet (**χ**^2^ = 7.01, *p* < 0.008, Table 1), indicating that where fish is cheaper, people consume more. These significant associations (per capita GDP, ranked price (inverted to reflect the negative association), total fish, and the percentage of marine fish available nationally) were developed into a four-point index, where a score of 4 indicated a greater probability of having a high dietary LCn-3. For those countries that had all four relevant pieces of data available, the distribution of the index scores did not differ if the country had an FBDG for seafood, had an FBDG but did not suggest a serving size, had an FBDG but did not recommend seafood, or did not have an FBDG (one-way ANOVA, F_3, 140_ = 1.70, *p* > 0.15, Figure 2). Overall, dietary LCn-3 increased in correlation with a lot of inexpensive fish of marine origin being available, along with a higher per capita GDP.

The United States is a country that not only includes fish in their FBDGs, but also have provided a serving size of 8 oz (226.7 g) per week (Dietary Guidelines Advisory Committee 2020). However, the typical American does not adhere to this level of consumption [9]. An estimated 90% of U.S. citizens do not meet the recommendation for seafood consumption [31]. In 2019, Americans ate an average of 138.9 g of seafood per week per capita [34]. This was less than the global average of 195.3 g/week per capita [20]. The top most consumed seafood item was shrimp, followed by salmon, then canned tuna (Figure 3). This high consumption of shrimp (30.5% of seafood intake) provides only 2.7% of the total average LCn-3s [42], whereas salmon (20% of consumption) provided a majority of the LCn-3s, at 62.9% (Figure 3). The general trends described here are based on recommendations for an adult eating a 2000-calorie diet, and do not consider the specific needs of pregnant women, infants, children, or those requiring a greater caloric intake. Not only do U.S. citizens not follow the dietary guidelines, but their preference for shrimp does not ensure that they obtain their necessary dietary LCn-3s.

## 4. Discussion

Here, the focus was on seafood contributions to dietary LCn-3s. The U.S. Dietary Guidelines identify seafood as “a protein foods subgroup that provides beneficial fatty acids (e.g., eicosapentaenoic acid [EPA] and docosahexaenoic acid [DHA])”. The recommendation of “eat 8 oz of seafood per week” in the United States (and elsewhere) is not being put into practice in a way that leads to adequate LCn-3 consumption. Although consumers are provided a message to “eat seafood”, they choose species that do not provide an adequate level of this critical nutrient. Aggregating seafood into the protein category in the FBDGs may limit the development of a more complete understanding of its nutrient benefits beyond protein. Just as low LCn-3 shrimp is selected as a seafood of choice, placing seafood within the protein category validates a substitution of other low-LCn-3 protein instead of seafood (“*instead of fish, I’ll have other animal source food…*”, Gardner, [6]). The same could be said of other nutrients, because seafood is beneficial for human health due to the contents of selenium, taurine, vitamins D and B12 [8], and can help to prevent deficiencies of these nutrients [3]. Nutrient-density is identified as important within the U.S. Dietary Guidelines (2020), being mentioned 175 times, and seafood is specifically described as a nutrient-dense food [31]. However, salmon is mentioned specifically within the U.S. Dietary Guidelines only ten times, and just one of those mentions specifically addressed it as being a high-LCn-3 food. Salmon is also identified once each as a source of calcium and Vitamin D, whereas it was mentioned eight times for having low methylmercury (one mercury mention also in the same reference to salmon being high in LCn-3s). Sardines and anchovies, two other fish high in LCn-3s, are not associated with this fact in the U.S. Dietary Guidelines, but instead are referred to as being seafood low in methylmercury (eight instances each, whereas sardines are highlighted once as being a good source of calcium). The U.S. Dietary Guidelines steer a consumer towards eating more healthily, although do not result in wiser consumption with respect to seafood. This is also observed globally, where countries with FBDGs for seafood have marginally but not statistically increased dietary LCn-3 consumption. FBDGs have a monumental task to encourage citizens to eat properly, but the overly general and misplaced guidance regarding seafood consumption (e.g., highlighting mercury over LCN-3s, Spiller et al., [10]) limits its outcome-based impact. This misplaced guidance is demonstrated through the U.S. FBDGs, where “omega-3” is mentioned once, “EPA/DHA” is mentioned four times, and methylmercury is mentioned 34 times. Furthermore, the ratio of omega-3 to omega-6 fatty acids is important for bioavailability, due to the shift to more omega-6 consumed in Western diets [43], and is not mentioned in the U.S. FBDGs. The FBDGs need to be explicit about solutions for meeting recommended nutrient intake, and need to include the species of fish, as well as strategies to improve the omega-3:omega-6 ratio. 

Increased LCn-3 consumption needs to align with social and environmental sustainability goals [44], and an environmentally sustainable consumption [45] approach to FBDGs is necessary, although has not yet been achieved [26]. Hallstrom et al. [32] have identified key species for Sweden that meet both high nutrient density and low environmental impact targets, and this work needs to be conducted internationally to identify the best sources to meet nutrient goals for human health. The most critical step to bridge this nutrient consumption gap is to adjust FBDGs to more specifically identify high-LCn-3 fish species. There are multiple non-animal proteins that have lower environmental impacts than seafood [46]; therefore, seafood may be overlooked from a protein standpoint, with consumers missing out on the LCn-3 benefits. Eating small, oily, pelagic fish will be important in meeting an increased LCn-3 consumption goal, as will salmon (and trout and char). There are multiple production platforms for salmon, each with a complementary suite of benefits and drawbacks from a life cycle assessment standpoint [47], and caution needs to be exercised in not blindly categorizing all aquaculture fish into high or low impact. Additionally, equity needs to be considered. Within the United States, there is regional variation in the availability of fish at chain-restaurants [22], and affordability needs to be a driver for adequate nutrition. However, the biologically relevant consumption of LCn-3s may only come through addressing global LCn-3 deficit [18] through the increased production of alternatives [48] such as algal oils, as well as preserving more of the nutrients in processing and the value chain [49], and finally, by aligning nutrition goals to policies [50]. If alternatives are to be created, the efficacy of LCn-3 consumption from supplements vs whole fish will need further examination. Studies of heart health mainly focused on LCn-3 supplements do not find overwhelming benefits [51], whereas studies examining whole fish consumption often do [52].

## Figures and Tables

**Figure 1 foods-10-01816-f001:**
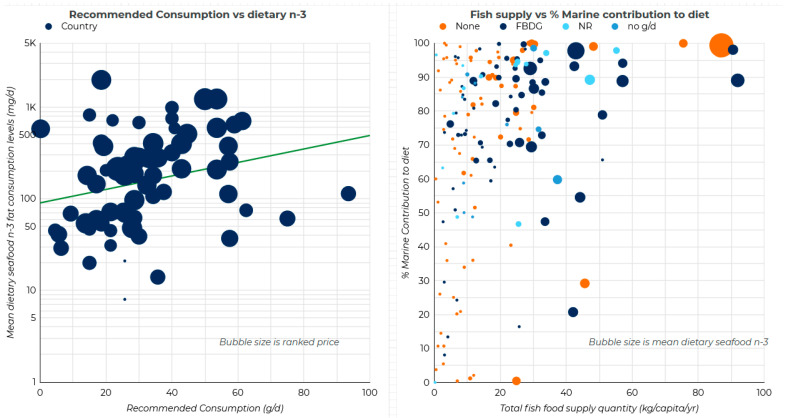
(**Left**) The comparison between the recommended consumption (g/d, [27]) and mean dietary seafood LCn-3 consumption [20], with bubble size indicating the ranked price [29]; (**right**) the total fish supply vs. the proportion of that which is of marine origin [28], with a bubble size of mean dietary seafood LCn-3 [20]. Interactive data are available at https://www.tlusty.solutions/n3.html (accessed on 28 June 2021).

**Figure 2 foods-10-01816-f002:**
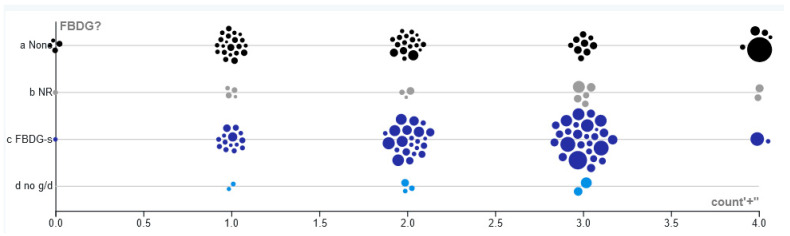
The number of countries that have 0–4 positive attributes for seafood consumption: GDP [38], ranked fish price [29], and total fish and percentage of marine fish available [28]. Positive attributes indicate that the country is above the median, except in the case of ranked fish price, which has a negative association with dietary LCn-3, and thus is inverse. The countries are split into (**a**) those that do not have an FBDG (None); (**b**) those that have no seafood recommendation in their FBDG (NR); (**c**) those that have a seafood recommendation in g/d (FBDG-s); and (**d**) those that have a seafood recommendation, but g/d is not specified (no g/d). Size of the bubbles indicates the country-level LCn-3 consumption [20].

**Figure 3 foods-10-01816-f003:**
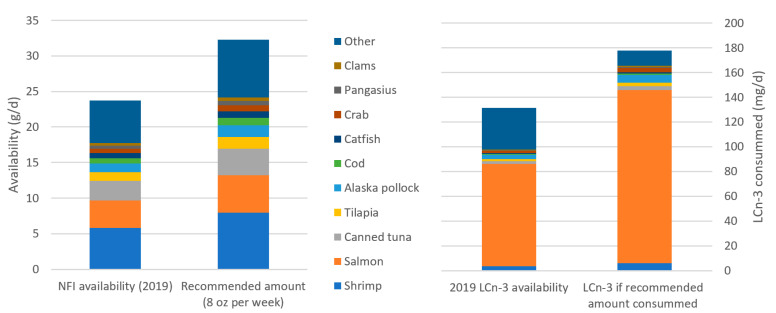
(**Left**) The availability of seafood species for American consumers in 2019 [34] and the values of each of those species if Americans consumed an aggregate 8 oz per week [31]. (**Right**) Dietary LCn-3 [42] for Americans based on seafood species available in 2019, and if they were to consume the recommended amount of seafood.

**Table 1 foods-10-01816-t001:** The association between dietary LCn-3 [20] and factors that may influence seafood intake. Association is determined by a median-quadrant approach, where data are assessed for each country if it is above or below the median value for that variable (in the case of FBDGs, it is the presence/absence of a seafood recommendation). A positive association is then determined if dietary LCn-3 and the other factors have a majority of observations in the below–below and above–above quadrant. Darker colors indicate stronger associations, lighter colors indicate weaker associations, and the numbers indicate a count of countries in each quadrant. The additional factors are recommended (g/d) intake and if the FBDG identified seafood [27], as well as GDP [38], ranked fish price [29], and total fish and percentage of marine fish available [28]. Price had a negative association (lower price associated with greater dietary LCn-3), whereas all other variables had a positive association.

		Dietary	N-3
		Below	Above
**Recommended**	Above	5	26
**g/d**	Below	19	15
**FBDG**	Above	36	56
	Below	47	27
**GDP**	Above	29	51
	Below	49	31
**Price**	Above	44	29
	Below	28	45
**Total fish**	Above	16	67
	Below	67	16
**% Marine Fish**	Above	31	52
	Below	52	31

## Data Availability

Interactive data are available at https://www.tlusty.solutions/n3.html (accessed on 28 June 2021).

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
