# Peer review of "Food-Based Dietary Guidelines for Seafood Do Not Translate into Increased Long-Chain Omega-3 Levels in the Diet for U.S. Consumers"

_foods, 2021, doi:10.3390/foods10081816_

Round 1

Reviewer 1 Report

Manuscript "Food Based Dietary Guidelines for seafood do not translate 2 into increased long chain omega-3 levels in diet" demonstrates an adequate understanding of the relevant literature in the field and cite an appropriate range of literature sources. The paper's argument is built on an appropriate base of theory and available statistical data. The results are presented clearly and discussed appropriately. Although the main finding is relevant for the US, it provides enough other results of interest for international readers.  The manuscript needs:

  1. minor editing (lines 35-36)
  2. clarification on the daily recommended nutritional intake of LCn-3 stated (lines 47-48) . In my knowledge it is 250-500 mg/day. Could you explain the value of 1000 mg? 

Author Response

Thank you for these comments.

My edits are:

Line 35-36:

original: Within high-income countries, people are still hungry, and those that aren’t, often do not get all their recommended nutrients [7]. It is important that substitutions be within the proper an intended purpose.

revised: Within high-income countries, people are still hungry, while for many of those that can acquire sufficient calories, they often do not get a full complement of recommended nutrients [7]. It is important that suggested nutritional substitutions be within the proper an intended purpose for achieving nutritional sufficiency.

In line 47, there are guidelines that go up to 1,000mg although this is limited, and so I edited this line to be "Most national dietary guidelines that have a  daily recommended nutritional intake of LCn-3 place the value at > 250mg per day "

Reviewer 2 Report

In the work “Food Based Dietary Guidelines for seafood do not translate into increased long chain omega-3 levels in diet” the author studied the association between dietary omega-3 consumption, and food-based dietary guidelines, gross domestic product, the ranked price of fish, and the total amount on the proportion of marine fish available.

The manuscript is very well written and easy to follow. However, some minor comments should be addressed:

  • Please improve Table 1 for a better understanding
  • Include a conclusion with some potential solutions/guidelines to promote an increase in the consumption of long-chain omega-3

Author Response

Thank you for the comments.

to improve the clarity of Table 1, I explain the colors to help further explain the table (Darker colors indicate stronger associations, while lighter colors indicate weaker associations, and number indicates a count of countries in each quadrant.), and also provide a guide or outcomes (Price has a negative association (lower price associates with greater dietary LCn-3), while all other variables have a positive association.).

Making solutions clearer was completed by highlighting some of the various options. This is rooted in behavior change by consumers and is going to be a slow ship to move. I made recommendations more explicit, and added the sentence "The FBDGs need to be explicit about solutions for meeting recommended nutrient intake, and need to include species of fish, as well as strategies to improve the omega-3:6 ratio."